# Acoustic enrichment can enhance fish community development on degraded coral reef habitat

Timothy A.C. Gordon[1,2]*, Andrew N. Radford[3], Isla K. Davidson[3], Kasey Barnes[4], Kieran McCloskey[1], Sophie L. Nedelec[1], Mark G. Meekan[2], Mark I. McCormick [4,5] & Stephen D. Simpson [1]

Coral reefs worldwide are increasingly damaged by anthropogenic stressors, necessitating novel approaches for their management. Maintaining healthy fish communities counteracts reef degradation, but degraded reefs smell and sound less attractive to settlement-stage fishes than their healthy states. Here, using a six-week field experiment, we demonstrate that playback of healthy reef sound can increase fish settlement and retention to degraded habitat. We compare fish community development on acoustically enriched coral-rubble patch reefs with acoustically unmanipulated controls. Acoustic enrichment enhances fish community development across all major trophic guilds, with a doubling in overall abundance and 50% greater species richness. If combined with active habitat restoration and effective conservation measures, rebuilding fish communities in this manner might accelerate ecosystem recovery at multiple spatial and temporal scales. Acoustic enrichment shows promise as a novel tool for the active management of degraded coral reefs.

[1] Biosciences, University of Exeter, Hatherly Laboratories, Prince of Wales Road, Exeter EX4 4PS, UK. [2] Australian Institute of Marine Science, Perth, WA 6009, Australia. [3] School of Biological Sciences, University of Bristol, 24 Tyndall Avenue, Bristol BS8 1TQ, UK. [4] Department of Marine Biology and Aquaculture, James Cook University, Townsville QLD 4811, Australia. [5] Australian Research Council Centre of Excellence for Coral Reef Studies, James Cook University, Townsville QLD 4811, Australia. *email: tg333@exeter.ac.uk

Climate change and local anthropogenic stressors are causing unprecedented damage to coral reefs globally[1,2], necessitating novel techniques to counteract degradation and proactively manage these rapidly changing ecosystems[3–5]. Taking active steps to maintain healthy fish communities will be vital in reversing reef degradation; fishes perform a diverse suite of important functional processes[6,7], meaning that damaged reefs have a higher chance of recovery if they have healthy fish populations[8,9]. Reef fish populations are sustained by recruitment, whereby young fish that spend their larval stage in the open ocean use a range of sensory cues to detect, orient toward, and settle to reef habitat[10,11]. However, degraded reefs smell and sound less attractive to juvenile fishes[12,13], and receive lower levels of fish settlement than healthy systems[14,15]. Artificially reversing degradation-associated sensory changes might restore habitat attractiveness, promote the settlement and retention of functionally important fish species, and enhance local-scale recovery processes. Acoustic cues are particularly amenable to artificial restoration, due to their use by a wide range of settlement-stage fishes[16,17] and their ease of manipulation in field conditions using underwater loudspeakers[13,16,17].

Here, we investigate acoustic enrichment as a novel management tool for aiding the mitigating and reversal of coral reef degradation. We use loudspeakers to broadcast healthy soundscapes on experimental coral-rubble patch reefs for 40 days during a natural recruitment season (November–December 2017) on Australia's northern Great Barrier Reef. We compare the developing fish communities on these acoustically enriched reefs with those on two categories of acoustically unmanipulated control reefs (with and without dummy loudspeaker rigs). We find that acoustic enrichment enhances fish community development within an important reef fish family, across a range of specific trophic guilds and at the level of the whole community. Rebuilding fish communities in this manner shows promise as a novel tool that might complement existing techniques for the active management of degraded coral reefs.

## Results

**Effects of acoustic enrichment on damselfish recruitment.** Acoustic enrichment had a significant positive impact on juvenile fish recruitment throughout the study period. Juvenile pomacentrids (damselfish that had settled in the current season on reefs following a pelagic larval stage) were repeatedly surveyed across 6 weeks; this family was chosen because they are non-cryptic, highly abundant (up to 50% of reef fish communities[18])

and individuals can be visually surveyed accurately with minimal disturbance to the developing fish community. Compared to both no-loudspeaker reefs and dummy-loudspeaker reefs, acoustically enriched reefs attracted damselfishes at a faster rate in the early stages of the experiment and maintained higher abundance throughout the 40 days (Fig. 1a). After 40 days, there were twice as many juvenile damselfishes on acoustically enriched reefs than both categories of acoustically unmanipulated reefs, with no significant difference between the two control treatments (generalised linear mixed model (GLMM), loudspeaker treatment: $\chi^2 = 54.732$, df = 2, $p < 0.001$; Fig. 1b; full model and post-hoc pairwise comparisons in Supplementary Table 1).

The asymptotic trajectories of damselfish abundance on reefs in the second half of the experiment (Fig. 1a) likely represent a stable dynamic equilibrium between settlement and predation, rather than a static population, for two reasons. First, the patch reefs were deployed asynchronously (i.e. some reefs started and finished their 40-day experimental period 10 days apart from others). This suggests that population stabilisation was due to population dynamics within each individual system rather than date-linked external factors, such as a region-wide change in currents reducing settlement levels in the later phase of the experiment. Second, direct predation of juvenile fishes by a range of taxa (e.g. *Carangidae*, *Pseudochromidae* and *Synodontidae*) was occasionally observed during surveys in the later stages of the experiment (T.A.C.G. pers. obs.); this is consistent with other studies that document high natural mortality rates of juvenile coral reef fishes[19,20]. An absence of overall declines in abundance during this phase of the experiment therefore suggests that ongoing recruitment was compensating for density-dependent predation of juvenile fishes.

**Effects of acoustic enrichment on fish community development.** Acoustic enrichment increased abundance of juvenile fishes across all major trophic guilds. Comprehensive whole-community surveys after 40 days revealed that there were significantly more herbivores, omnivores, planktivores, invertivores and piscivores on acoustically enriched reefs than on acoustically unmanipulated reefs, with no significant differences between the two control groups (dummy-loudspeaker and no-loudspeaker) in four of the five trophic guilds (linear mixed models (LMMs) and GLMMs, loudspeaker treatment: $\chi^2 = 7.499$–43.473, df = 2, $p = <0.001$–0.024 (Fig. 2); full models and post-hoc pairwise comparisons in Supplementary Table 1). Attraction and settlement of a range of trophic guilds is important because

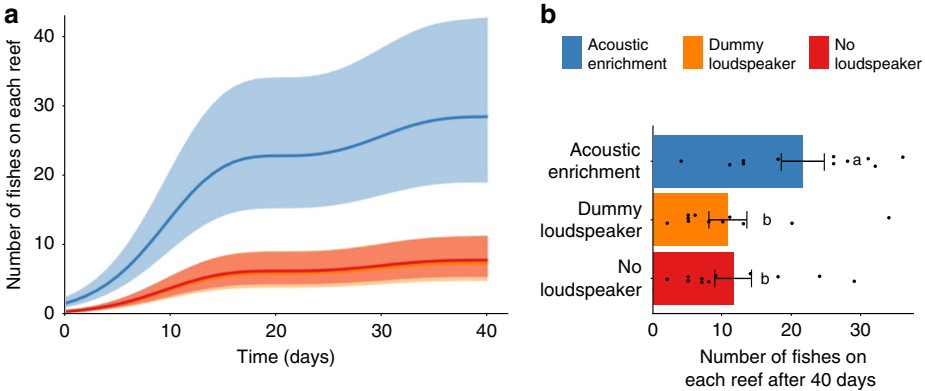

**Fig. 1** Effect of acoustic enrichment on damselfish community development. **a** Outputs from a generalised additive mixed model, modelling number of juvenile damselfish per reef (mean ± SE) over time based on repeated surveys; orange and red lines and ribbons are almost completely overlapping. **b** Raw count data of number of juvenile damselfish per reef (mean ± SE) from surveys undertaken after 40 days. Different letters associated with bars represent significant differences in post-hoc Tukey's HSD tests, following a significant effect of loudspeaker treatment in a generalised linear mixed model (see Supplementary Table 1).

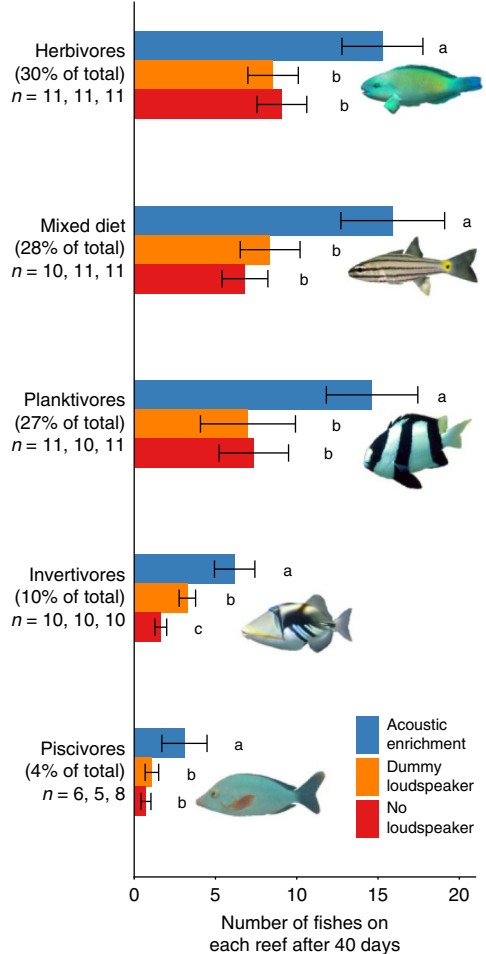

**Fig. 2** Effects of acoustic enrichment on different trophic groups. Mean ± SE juvenile fish abundance in different trophic guilds on experimental patch reefs. Y-axis labels give the proportion of all fishes and the frequency of occurrence (number of populated reefs in each loudspeaker treatment) represented by each trophic guild. Mixed-effects models revealed significant effects of loudspeaker treatment in all five trophic guilds; different letters associated with boxplots represent significant differences in post-hoc Tukey's HSD tests (Supplementary Table 1). Images of fish are taken from the Lizard Island Field Guide (lifg.australianmuseum.net.au), licensed under the Creative Commons Attribution 3.0 License (creativecommons.org/licenses/by/3.0/).

assemblages of coral reef fishes are inherently high in functional diversity[7], with the functional contributions of many species likely to be underappreciated[21].

Consistent differences across multiple trophic guilds resulted in a positive effect of acoustic enrichment at the community level. Acoustically enriched reefs had twice the total abundance of juvenile fishes of dummy-loudspeaker and no-loudspeaker reefs, which were not significantly different from each other (LMM, loudspeaker treatment: $\chi^2 = 21.107$, df = 2, $p < 0.001$; Fig. 3a; full model and post-hoc pairwise comparisons in Supplementary Table 1). Acoustically enriched reefs also had 50% greater species richness than dummy-loudspeaker and no-loudspeaker reefs, with no significant difference between the two acoustically unmanipulated treatments ($\chi^2 = 12.848$, df = 2, $p = 0.002$; Fig. 3b; Supplementary Table 1). Further, there was a significant effect of loudspeaker treatment on effective Shannon diversity, with acoustically enriched reefs having greater diversity than reefs with no loudspeaker; dummy-loudspeaker reefs were not

significantly different to either of the other two treatments ($\chi^2 = 5.990$, df = 2, $p = 0.050$; Fig. 3c; Supplementary Table 1). This development of a more abundant, more species-rich and more diverse fish community is potentially the result of increased detectability of acoustically enriched reefs (i.e. fishes could hear, and therefore detect and orient towards, reefs from a greater distance away), or altered settlement behaviour of juvenile fishes in response to additional sounds (i.e. fishes were more likely to settle onto the reef once they arrived at it), or a combination of both mechanisms.

## Discussion

Our results demonstrate that acoustic enrichment has the potential to enhance community development of fishes on degraded coral reef habitat. Playback of healthy reef sound: (a) created temporally stable population increases in the most abundant taxonomic group of reef fishes (Pomacentridae; Fig. 1); (b) drove increases in fish recruitment across a broad range of trophic guilds (Fig. 2); and (c) led to increased abundance, species richness and diversity at the whole-community level (Fig. 3). The near-ubiquitous qualitative equivalence of communities associated with dummy-loudspeaker and no-loudspeaker control treatments demonstrates that observed increases on acoustically enriched reefs were due to the acoustic treatment, rather than the additional visual cues or structural complexity associated with the presence of loudspeakers. The current work did not attempt to investigate which sounds are most effective at attracting settlement-stage fishes. However, previous evidence from short-term trials shows that healthy reef sound is more attractive than either degraded reef sound or white noise to settlement-stage fishes[13,22], suggesting that the results of the current study are likely influenced by more than indiscriminate phonotaxis.

Acoustic enrichment may have effects on community dynamics at multiple spatial and temporal scales. First, many fishes migrate away from their initial settlement site during ontogeny[23]. As such, even though loudspeakers inevitably create smaller acoustic halos than natural reefs, elevated settlement to acoustically enriched nucleus sites may promote fish community development at wider spatial scales. Additionally, initial increases in settlement driven by acoustic enrichment may facilitate a 'snowball effect', whereby other fishes respond positively to communities established earlier, causing further increases in settlement[24,25]. Finally, this community development might also cause natural soundscapes to increase in volume, because healthy reef ecosystems are louder and more acoustically complex than their degraded counterparts[13]. These mechanisms all raise the possibility that the effects of acoustic enrichment could extend beyond those observed in the immediate vicinity of loudspeakers during their deployment, with benefits seen at larger spatial and temporal scales (Fig. 4).

Acoustic enrichment shows potential as a sensory-based conservation tool for contributing to the restoration of coral reef ecosystems. Trials of sound playback in terrestrial contexts have previously revealed its potential to alter animal behaviour and increase settlement rates of acoustically specialised taxa[26–28]. We show that a similar technique exhibits promise as a novel conservation tool for the management of degraded coral reefs. Existing management and restoration techniques can improve habitat quality in previously degraded areas[3,29,30]; if combined with such techniques, acoustic enrichment might accelerate fish community development and enhance natural ecosystem recovery processes. Further work is now needed to investigate the translatability of this finding into different reef habitats and geographical contexts; the impacts of acoustic enrichment on adult fish behaviour; the long-term recovery of natural settlement

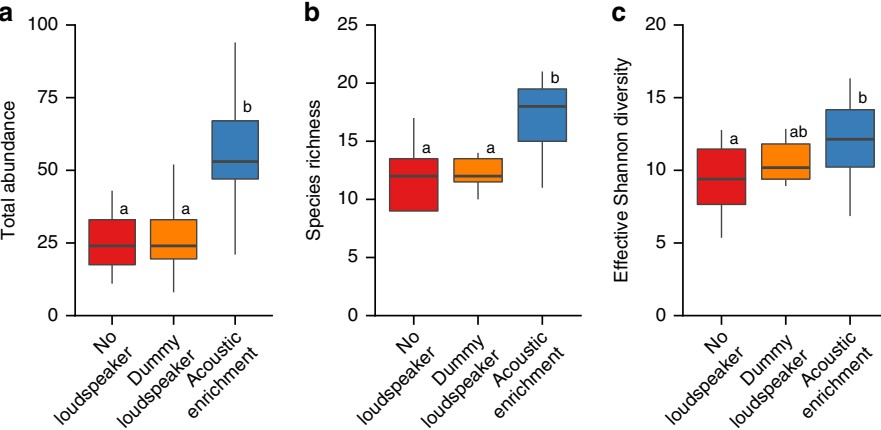

**Fig. 3** Community-level effects of acoustic enrichment. **a** Total abundance, **b** species richness, and **c** effective Shannon diversity of juvenile fish communities on experimental patch reefs. Boxplots represent medians (central lines), interquartile ranges (boxes) and 95% ranges (whiskers). Linear mixed models revealed significant effects of loudspeaker treatment in all three cases; different letters above boxplots represent significant differences in post-hoc Tukey's HSD tests (Supplementary Table 1). Absolute values for species richness and Shannon diversity are likely to be underestimates, as fish that were only identified to higher taxonomic levels (family or subfamily) may represent more than a single species.

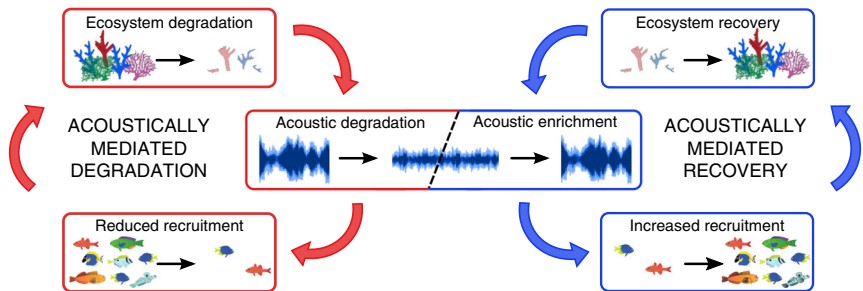

**Fig. 4** Schematic illustrating the potential for acoustic enrichment to reverse harmful feedback loops on coral reefs. The left-hand cycle shows acoustically mediated feedback associated with degradation[13]; the right-hand reverse cycle suggests how acoustic enrichment could facilitate ecosystem recovery through increasing recruitment and enhancing communities of fishes. Pictures of fishes are adapted from ref. [12].

cues; and the spatial scale of effects on fish communities and ecosystem processes. This will optimise acoustic enrichment as one of a suite of active management tools for restoring degraded coral reef ecosystems.

## Methods
**Study site**. This study was carried out during October–December 2017 in the lagoon to the south-west of Lizard Island Research Station (14°40.8′S, 145°26.4′E; Fig. 5). Lizard Island is a continental mid-shelf island in the northern Great Barrier Reef with an extensive surrounding fringing and lagoonal reef system. In the two years preceding this study, severe mass bleaching events caused extensive coral mortality in this area (over 60% of live coral bleached[31]), leading to widespread ecosystem change[32,33].

**Experimental design**. Prior to establishing experimental sites, the potential detection distance of reef-sound playback by juvenile fishes was determined using recordings at the study site. Recordings were taken from the deepest experimental reef (4.5 m mid-tide depth), to provide a conservative estimate of acoustic isolation; lower frequency (longer-wavelength) sounds attenuate relatively quickly in shallow water[34], so the potential detection distance of reef-sound playback is likely to have been higher here than at all other sites. Full-night recordings of reef-sound playback were taken simultaneously in both the sound-pressure and particle-acceleration domains, at 1, 50 and 100 m distance from a loudspeaker. Loudspeaker playback was conducted using the same methods as in subsequent experimental trials, as described below. Recordings were made in sea states between 0 and 2 on the Beaufort scale, and never during rain. Recording equipment was suspended 1 m above the seabed, hanging from a submerged stand to avoid unwanted noise from waves slapping on the hull of a surface vessel. Sound pressure was recorded using an omnidirectional hydrophone with inbuilt digital recorder (SoundTrap 300 STD; Ocean Instruments NZ, Auckland, New Zealand), and particle-acceleration recordings were taken using a triaxial accelerometer (M20–040; sensitivity following a curve over the frequency range 0–5 kHz; Geospectrum Technologies,

Dartmouth, Canada) connected to a digital 8-track recorder (F8 field recorder; Zoom Corporation, Tokyo, Japan). Recordings were all taken at a sampling frequency of 48 kHz, and analysed using the paPAM[34] (particle acceleration) and PAMGuide[35] (sound pressure) packages on MATLAB, across a frequency range of 10–4000 Hz as the likely hearing range of many juvenile fishes[36,37], with a Hamming window, a 50% overlap and an FFT size of 2048.

Analysis of recordings of reef-sound playback could not distinguish a signal against the background noise floor at a distance of 50 m. Each recording was analysed both as a full-night track and as a series of 20 time-matched 2-min sub-samples, evenly spaced throughout the night (Fig. 6). Recordings showed that loudspeaker playback recorded at a distance of 1 m had significantly increased received sound-pressure level (SPL) and particle-acceleration level (PAL) relative to playback recorded from both 50 and 100 m, with visible differences in power spectral density plots (Fig. 6). By contrast, recordings taken at 50 m from the loudspeaker had power spectra and received SPLs and PALs that were equivalent to those taken at 100 m (Fig. 6). Further, intermittent recordings taken at 50 m from a loudspeaker showed no significant difference in received SPLs and PALs within the hearing range of young fishes (10–4000 Hz) during 10-min periods of the night when the loudspeaker was turned on compared to equivalent 10-min periods when the loudspeaker was turned off (LMM, $n = 20$, SPL: $\chi^2 = 2.785$, df = 1, $p = 0.103$; PAL: $\chi^2 = 0.447$, df = 1, $p = 0.508$). Thus, recordings of reef-sound playback taken at a distance of 50 m were unable to distinguish the playback signal over the natural ambient conditions at the study site. Based on this finding, all experimental patch reefs were placed a minimum of 100 m from each other (Fig. 5), to achieve acoustic isolation of experimental sounds between reefs.

Experimental coral-rubble patch reefs were used to assess the impact of acoustic enrichment on fish community development. At the start of the fish recruitment season, 33 patch reefs were built on open sand, placed at a fixed distance of 25 m (as determined by GPS) from the nearest natural reef in 2–4.5 m water depth (mid-tide depth; tidal range during experiment ± 1.3 m). There was no significant difference in depth between the three treatment groups (linear model: $\chi^2 = 2.042$, df = 2, $p = 0.15$). Reefs consisted of 70 l of dead coral rubble, collected from a single degraded reef near the study site (Fig. 5), arranged around a double breeze block (40 × 40 × 20 cm) to create a structurally complex habitat patch of

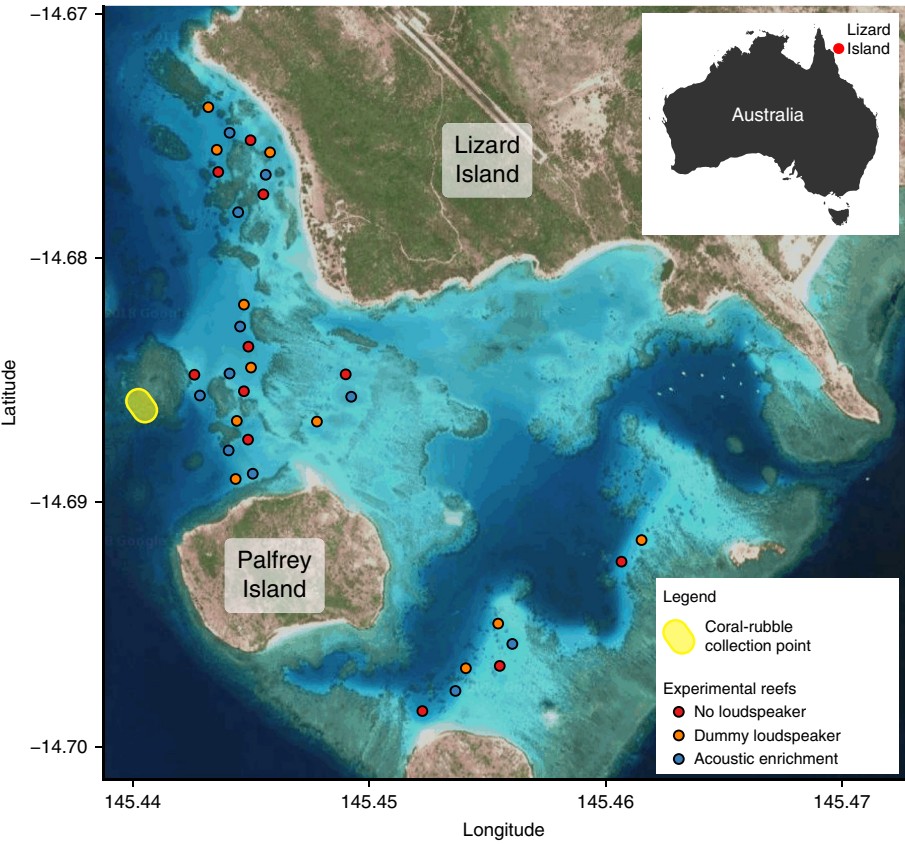

**Fig. 5** Study site map, showing experimental reefs and coral-rubble point. All experimental reefs were placed at a minimum distance of 100 m from their nearest neighbour, and at a fixed distance of 25 m from the nearest natural reef. Satellite image obtained from Google Maps, available at https://goo.gl/maps/5Loa45HKYgJ2 (Map data: Google, GBRMPA).

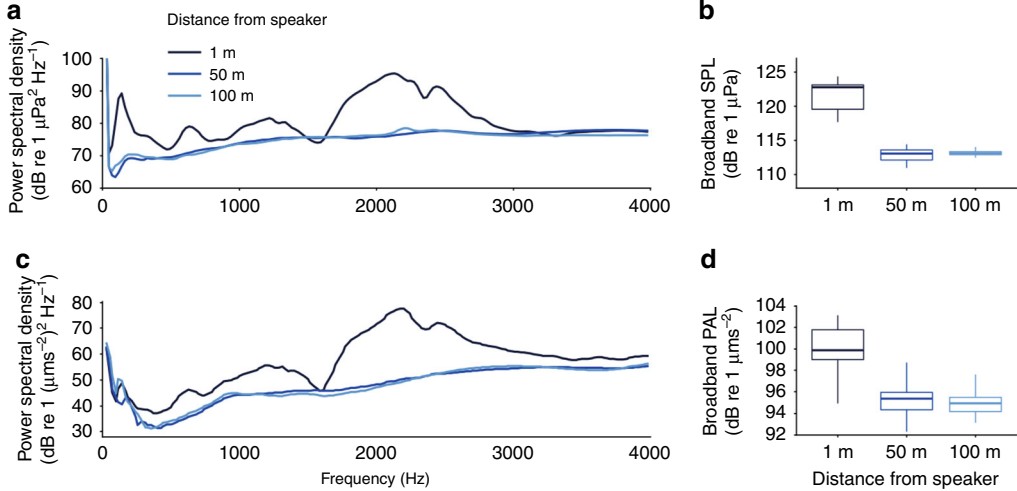

**Fig. 6** Recordings of reef-sound playback taken at 1, 50 and 100 m from the loudspeaker. Shown are: **a** power spectral densities from a single full-night recording taken in sound-pressure; **b** boxplots of total received broadband (10–4000 Hz) sound pressure level (SPL) from evenly spaced 2-min sub-samples throughout the night ($n = 20$); **c** power spectral densities similar to **a** but showing particle acceleration; and **d** boxplots of total received broadband (10–4000 Hz) particle-acceleration levels (PAL) similar to **b**. Boxplots in **b** and **d** represent medians (central lines), interquartile ranges (boxes), and 95% ranges (whiskers).

$2 \times 2 \times 0.5$ m. Each reef was composed of an approximately even volume of branching (60%), tabular (20%) and cuboidal/spherical (20%) rubble types. This created standardised patches of marginal reef habitat, to which many coral reef fish families have been seen to settle in earlier studies[13].

Reefs were assigned to one of three experimental treatments. Each reef was coupled with either no loudspeaker, a dummy loudspeaker system (to control for additional visual cues and structural complexity provided by loudspeaker systems), or a real loudspeaker system playing back healthy reef sound overnight (acoustic-enrichment treatment). Loudspeaker treatment was allocated pseudorandomly to reefs, avoiding allocation of the same treatment to reefs that were spatially adjacent. Acoustically enriched reefs were fitted with an underwater loudspeaker (University Sound UW-30; maximal output 156 dB and 1 μPa at 1 m, frequency response

0.1–10 kHz; Lubell Labs, Columbus, OH, USA) powered by an amplifier (M033N; 18 W, frequency response 0.04–20 kHz; Kemo Electronic GmbH, Germany) and a 12 V 12 Ah sealed lead-acid battery, connected to an MP3 player (Clip Jam; SanDisk, Milpitas, CA, USA) that was powered by an external battery pack (MIX IT RockStar, 10,000 mAh; Belkin, Los Angeles, CA, USA). Loudspeakers were fixed to the centre of reefs and oriented upwards, to ensure even distribution of sound in all directions laterally. Amplification systems were housed in a barrel floating directly above reefs, attached by a rope to a breeze block. Dummy-loudspeaker reefs consisted of a concrete tile of the same size, shape and colour as the loudspeakers, fixed to the reef in the same manner as the loudspeakers, attached by a rope to a floating buoy of the same dimensions as the barrel used in loudspeaker systems.

The acoustic-enrichment treatment consisted of playback of recordings taken at a healthy reef in the middle of the study site in November 2015; this was before the study site experienced two severe mass bleaching events in 2016 and 2017 that caused extensive coral mortality and widespread ecosystem change throughout the region[31–33]. Five different recordings were used to reduce pseudoreplication, with recordings being allocated to acoustically enriched reefs at random (each recording was allocated to 2–3 reefs). Recordings used in playback were full-night recordings, taken using an omnidirectional hydrophone (HiTech HTI-96-MIN with inbuilt preamplifier, manufacturer-calibrated sensitivity −164.3 dB re 1 V μPa$^{-1}$; frequency range 0.002–30 kHz; calibrated by manufacturers; High Tech, Inc.) connected to a digital recorder (PCM-M10, 48 kHz sampling rate; Sony Corporation). The hydrophone was freely suspended 1 m above the seabed in water depth of 3.5 m, from a rope-anchored barrel that contained the recorder.

Reef-sound playback was conducted overnight. Fish settlement is predominantly a nocturnal behaviour[38,39]; playback therefore started 0.5–1.5 h before sunset and stopped 2–5 h after sunrise the following morning. Playback was also matched by time of night to within 45 min of the original recording, such that playback tracks were playing approximately 'real-time' throughout the night. For an illustrative waveform and spectrogram of the reef-sound playback, see Supplementary Fig. 1. Equipment failure or rough weather conditions prevented loudspeaker deployment on some nights, but all acoustically enriched reefs had successful loudspeaker deployments on between 34 and 36 out of a total of 40 nights in the experiment.

**Surveys of fish communities**. The family Pomacentridae (damselfish) was surveyed regularly throughout reef deployment. The high abundance of damselfishes on coral reefs (up to 50% of reef fish communities[18]) facilitates adequate statistical power to test for differences in community development, and their non-cryptic nature allows accurate surveying with minimal disturbance. Visual surveys by a SCUBA diver (T.A.C.G.) were used to monitor communities of juvenile damselfishes (those that had settled on reefs following a pelagic larval phase in the current season). The observer and dive buddy remained at least 1 m from the reef during surveys, in order to minimise disturbance to the community. Each reef was surveyed 10 times throughout a 40-day period that started immediately following construction, with 3–9 days between consecutive surveys. The start of the survey period on each reef was staggered across a total duration of 10 days (i.e. construction and surveying of the final reef started 10 days after the first reef was built), to allow a single observer (T.A.C.G.) to complete surveys at the same experimental time point. Surveys took up to 1 h per reef; staggering was therefore necessary to allow for surveying of 33 patch reefs at the same experimental time points. Deployment order was counterbalanced such that the same number of reefs within each treatment were constructed and surveyed on each day.

After 40 days, the whole fish community on each reef was surveyed by dismantling the reef piece-by-piece. An observer using SCUBA (T.A.C.G.) checked each piece of rubble thoroughly, using dilute clove oil and a hand net to capture all juvenile fishes on the reef. The dive buddy kept a continuous watch for 'stray' fishes that attempted to escape across the sand flat or burrow into the sand during the survey process. All reefs were double-checked for missed fishes after surveys were completed; to our knowledge, no fishes were missed from surveys. All fishes were identified to species, except in cases where uncertainty meant that identification was only possible to family or sub-family level. In these cases, fishes that looked similar were assigned as the same species (e.g. 'Unknown goby 1'); this was the case for 4% of species, whose members together constituted 9% of the total abundance. Adult fishes were excluded from analyses, as their larger home ranges mean that they do not exhibit fixed associations with reef habitat to the same extent as juveniles[40]. After the experiment, all fish were released alive onto neighbouring reefs.

**Statistical analyses**. Visual surveys throughout the 40-day period were used to create accumulation curves for juvenile damselfishes (family Pomacentridae). A generalised additive mixed model (GAMM) with a negative binomial distribution included time (days since patch reef creation) as a smooth term, loudspeaker treatment (acoustic enrichment, dummy loudspeaker, no loudspeaker) and patch-reef ID as parametric coefficients, and playback-track ID as a random term. A first-order autoregressive structure was used to account for temporal autocorrelation, and visual examination of diagnostic plots and comparison of model outputs to raw data were used to confirm goodness-of-fit.

Comprehensive surveys at the end of the 40-day period were analysed using LMMs and Poisson-distributed GLMMs. Separate models were run for abundance of juvenile damselfishes (family Pomacentridae), herbivores, omnivores, planktivores, invertivores and piscivores, as well as total community abundance, species richness and effective Shannon diversity (calculated as the exponential of the Shannon–Weiner index[41]). Trophic classification was based on published literature[33,42–45] and FishBase[46]; for full details, see Supplementary Table 2. For fish that were only identified to family or sub-family, trophic guilds were assigned to all species within the relevant taxonomic group that are known to occur in the Lizard Island area (using lists compiled by the Lizard Island Field Guide [lifg.australianmuseum.net.au]), and the most commonly occurring guild in the taxonomic group was chosen as the trophic guild for that individual. Corallivores made up <0.25% of all fish, and were found on only two of 33 reefs, so were excluded from this analysis due to a lack of statistical power. All other trophic guilds represented at least 4% of the total count and were found on at least 19 of 33 reefs. In each model, playback-track ID was included as a random term, and error distributions (Gaussian LMM or Poisson-distributed GLMM) were chosen such that there were no deviations from homoscedasticity or normality in visual examinations of residual plots. The effect of loudspeaker treatment on the dependent variable was tested through ANOVA comparisons with a null model and post-hoc Tukey's HSD testing following statistically significant initial results ($p \leq 0.05$).

All figure creation and statistical modelling was conducted in R v. 3.5.0[47]. Figures were prepared using the packages cowplot[48], ggmap[49], and ggplot2[50]. Statistical modelling was conducted using the packages lme4[51] and mgcv[52].

**Ethical approval**. Permission and ethical approval for this work was granted by Lizard Island Research Station, the Great Barrier Reef Marine Park Authority (G13/35909.1), James Cook University (A2408, A2361), and the University of Exeter (2013/247).

**Reporting summary**. Further information on research design is available in the Nature Research Reporting Summary linked to this article.

## Data availability
Raw data are available from the University of Exeter's institutional repository at https://doi.org/10.24378/exe.1904.

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

## Acknowledgements

We thank the staff at Lizard Island Research Station for logistical support; Brendan Nedelec and Maggie Travis for fieldwork assistance; Howard Choat, Kendall Clements and William Robbins for assistance with fish identification; James Campbell for assistance with acoustic analyses; Danielle Dixson for drawings of fishes used in Fig. 4; and Patrick Kennedy, Harry Harding and Katherine Maltby for statistical advice. This work was supported by funding from a Natural Environment Research Council Research Grant NE/P001572/1 (to S.D.S. and A.N.R.); a Natural Environment Research Council-Australian Institute of Marine Science CASE GW4+ Studentship NE/L002434/1 (to T.A.C.G.); an Australian Research Council Discovery Grant DP170103372 (to M.I.M.); the Australian Institute of Marine Science (to M.G.M.); and the University of Exeter (to S.D.S.).

## Author contributions

T.A.C.G., A.N.R., S.L.N., M.G.M., M.I.M. and S.D.S. designed the research; T.A.C.G., I.K.D., K.B., K.M., S.L.N. and S.D.S. performed the fieldwork; T.A.C.G., A.N.R. and S.D.S. analysed the data; T.A.C.G. wrote the initial manuscript and all authors contributed to its final revision.

## Competing interests

The authors declare no competing interests.
