## [Peer Review File · Nature Communications]

Reviewers' Comments:

Reviewer #1:

Remarks to the Author:

This is a good paper. I think it requires some revision in methods/data analyses, as well as wording, and perhaps in the next round will be ready for publication.

1. I think the authors need to be more cautious when they refer to 'restoration'. They have created an acoustic playback treatment, similar to visual treatments such as duck decoys or bird call boxes to attract birds. They are not restoring the habitat. They are simulating an attractive environment. New words should be chosen. Playback. Or Healthy reef or reef sounds. Or Treatment. Not of this takes away from the results, but allows the data to be presented accurately, and impartially. This word is found in the title and many places. Please remove and change it with something less partial.

2. You are not actually restoring a reef, or a soundscape. If you worked from a degraded reef, that would be different, but new coral rubble is not a place many of these spp would normally settle.

Line 55 – again, 'restoration'. Change this word out.

Line 69-71 – why were these deployed and started on different days? How? Was there an order? Was start tested in the GLM? Changes may be suggested

Line 97-98 – 'Solutions' sentence is discussion, not a result. Please delete.

Lines 182-183 – This paper cited here exists, but it is not necessarily the fully story. See work by Jamie McWilliams and Miles Parsons, using longer-duration data, where bleaching occurs on the GBR, but the soundscapes were not radically different before and after. Further, the cited paper does not have enough data to make this claim. Further, why is the statement needed? Pls delete it.

Line 192 – I am fully confused by the terms juvenile, recruit, and settlement. What age class of fish are you referring to in this work? Is it consistent for all taxa studied? Please define your terms and keep one consistent term throughout the text.

Line 194-196 – what does this mean 'Loudspeaker playback matched that used in subsequent experimental trials' ? The sound levels and spectra matched the healthy ecosystem? Then show the healthy system and the playback plotted together. This is a poorly worded and imprecise statement. The methods were identical but stimuli were not measured in each location?

Also, you noted the bleaching impacted this system, so are you using impacted sounds? i.e., a poor soundscape for your treatment?

Later in this paragraph – So what were the depths at each site? This really affects sound transmission and sound levels.

- How was the Geospectrum particle motion data analyzed? This is not trivial? Sample rates? Was there a lowpass filter? Analyses window? Overlap?

Line 207 – 'Analysis suggested....' What analyses? Describe your methods. Sorry, this is very incomplete. Further, inaudible is not the right word here. Do you mean you could not distinguish the signal from the background noise? Note that in masking, temporal components of a signal play a huge roll. What was your integration time of analyses?

Line 210 – what does 'characteristically altered power spectra' mean? Please be clear. This is not very precise. Also delete volume and used SPL or particle acceleration or received level.

Line 212 – particle acceleration units are not very helpful or standard. See Casper and Mann papers, or Karlsen and Sand papers. Usually it is reported in m/s² or dB m/s but i guess it is because the differences are so small here (micrometers). Is that a realistic value? What is your variance?

Line 213 – again, volume is not a scientific word. You may be working in intensity or amplitude...

216 – what you testing here, full band SPL or certain bands?

219 – Masking is a hearing question. Inaudible is not correct. Revise this. See work by J. Stanley and Richard Fay to properly assess what can and cannot be detected. This assumption needs to be revised.

251 – it is cleaner to play a silent treatment in case there is an electromagnetic signal of the system.
262 – 1 m from surface but what was the depth (distance from the sound source)?
271 – did you measure the received levels during the playback experiments? At least from a subset of your trials? Or at least calibrate each site? Measuring and monitoring your treatment is crucial.
296 – did you test distance to nearest reef? Seem like an important parameter. Add if not.
293 – what do believe your catch success is/was? Please report. This is a difficult job, catching these fish.
Additional questions- where are you getting the rubble? From the beach, from near the site they were placed? All From one location?

Figure 1 – You mention there was an increase relative to the noon-moon/time of greatest recruitment. But also that there was a 10-day spread relative to treatment start date. Due to start-date-spread, it is not very easy to see the lunar affect, although this is referred to in the text. Can you plot the data relative to new-moon? Also, What does 'abundance' mean? Number of animals per site? 'Raw data' is not very explanatory... Suggest phrasing the x-axis of b or defining in the caption.

Figure 4 – Delete this figure. It's confusing. I know generally what you want to show but I don't think it is represents what you want to show. For example, on the blue side, it looks like the recovered ecosystem is producing the low level sound. but really, it is more linear with increased sound leading to recruitment and recovery. I get the goal, but I don't think it's well devised.

Figure 5 – Lizard Island seems incorrectly placed on mainland Australia. Note, it is off shore. Pls correct.

Overall, What are the currents in the area? Were they measured? Did you see drop-off of recruitment based upon sites?

Reviewer #2:

Remarks to the Author:

This study eloquently shows that by recording healthy reefs (in this case from before recent local bleaching events) and broadcasting this sound on degraded experimental patch reefs through underwater speakers, juvenile reef fishes settle in higher abundance and diversity compared to those without the broadcast. This is important given the current and projected state of coral reefs and the impacts reef degradation has on fish communities. This study showed that fish abundance increased across a range of functional groups by the completion of the 40 day experiment when the recording was broadcast from the reefs. Based on these results, the authors suggest that this method could be used as a restoration tool to help restore fish populations on degraded reefs. While this study is well executed, presented, and analyzed, the findings that broadcasted reef sound increases juvenile fish abundance has been previously demonstrated and published by some authors in this study. My concern with the implications of using sound as a tool to recover fish populations (and I acknowledge that you also state "complementing existing local-scale reef management and restoration projects" in the conclusions) is that while the emitted sound might promise the fish a healthy environment to settle, it does not provide the essential resources that a healthy environment offers. Therefore, it attracts them to the reef under false pretenses. Reef fishes are attracted to these sounds because they are cues for a suitable place to settle. Therefore, this methods may fool fish to settling into an environment that may not facilitate survival. While this study shows a higher abundance over the 40 days sampling period, it is unclear if higher recruitment is masking high mortality or if there are long term implications for settling onto a reef devoid of resources. This may be species specific also, whereby fish abundance and diversity increases, but the post-disturbance community is different to

the pre-disturbance community. I understand that this is outside the scope of this study, but long term health and survivorship of species is important to fully understand if this method is a useful tool and therefore can facilitate recovery. Given the sound is only distinguishable at a close distance (<50m), it would appear that this method is not attracting them to the reef, rather it is influencing them to settle on the reef as they pass by. Tricking fishes to settle in a degraded habitat might be a death sentence, and it might be better to let these fish continue in the water column until they find a suitable habitat for settlement (in the case of patchy degradation). Fish communities change following a decline in reef healthy because it no longer provides resources for these fishes. Therefore, it is difficult to make the case that attracting fish through recruitment will recover fish populations if the environment is the same as when the fish declined. Aside from these two points, the authors have conducted an excellent study that is well presented and executed, and makes a valuable contribution to the field.

Response to reviewers: NCOMMS-19-01808: “Acoustic enrichment can enhance fish community development on degraded coral-reef habitat”

We are very grateful to both reviewers for their detailed, positive and constructive comments.

Below, we address each of the reviewers’ comments in turn. We present the original comments as *italicised text*, followed by our response to each comment in **bold**. Line numbers refer to those in our revised manuscript, where all modified and referenced sections are **highlighted in yellow**. In our responses, we have only included comments that raise questions or require action; reviewer comments that were exclusively complimentary and/or summary statements have been left out of this response document.

Reviewer 1

Comment 1.1: I think the authors need to be more cautious when they refer to ‘restoration’. They have created an acoustic playback treatment, similar to visual treatments such as duck decoys or bird call boxes to attract birds. They are not restoring the habitat. They are simulating an attractive environment. New words should be chosen. Playback. Or Healthy reef or reef sounds. Or Treatment. Not of this takes away from the results, but allows the data to be presented accurately, and impartially. This word is found in the title and many places. Please remove and change it with something less partial. You are not actually restoring a reef, or a soundscape. If you worked from a degraded reef, that would be different, but new coral rubble is not a place many of these spp would normally settle.

Response 1.1: We have replaced all instances of ‘soundscape restoration’ and its grammatical equivalents with ‘acoustic enrichment’ and its grammatical equivalents. We have done this throughout the document: in the title (line 1); the abstract (lines 24, 25 and 29); the main text (lines 47ff.); the figures; and the methods (lines 233ff.).

Comment 1.2: Line 55 – again, ‘restoration’. Change this word out.

Response 1.2: This word has been changed (see response 1.1).

Comment 1.3: Line 69-71 – why were these deployed and started on different days? How? Was there an order? Was start tested in the GLM? Changes may be suggested.

Response 1.3: Reefs were deployed on different days so that all subsequent fish surveys could be completed by the same observer at the same experimental time point. Accurate surveying of the small and, in some cases, cryptic fishes found on the experimental reefs can take considerable time (up to 1 hour per reef); it would be impossible for the same observer to survey 33 patch reefs accurately in a single day. In order to have all experimental reefs surveyed on the 40th day of their deployment, it was therefore necessary to stagger the start dates over a 10-day period. Reef deployment was counterbalanced such that the same number of reefs within each experimental treatment group were deployed on each day; as such, start date was not included in statistical models. All these details have been added to the methods (lines 288–292).

Comment 1.4: Line 97-98 – ‘Solutions’ sentence is discussion, not a result. Please delete.

Response 1.4: This has been deleted.

Comment 1.5: Lines 182-183 – This paper cited here exists, but it is not necessarily the fully story. See work by Jamie McWilliams and Miles Parsons, using longer-duration data, where bleaching occurs on the GBR, but the soundscapes were not radically different before and after. Further, the cited paper does not have enough data to make this claim. Further, why is the statement needed? Pls delete it.

Response 1.5: This has been deleted.

Comment 1.6: Line 192 – I am fully confused by the terms juvenile, recruit, and settlement. What age class of fish are you referring to in this work? Is it consistent for all taxa studied? Please define your terms and keep one consistent term throughout the text.

Response 1.6: ‘Recruit’ (noun) has been removed entirely from the manuscript. ‘Recruitment’ (verb) refers to the act of young fish detecting, orienting towards, and settling to reef habitat after their pelagic larval phase; this is now defined as such in the first paragraph of the main text (lines 38–40) and used consistently throughout the manuscript. ‘Settlement’ is used exclusively to describe the arrival stage of the recruitment process, and is now used consistently as such throughout. ‘Juvenile’ refers to all fishes that are in their first season (i.e. they have arrived at and settled on the reef for the first time this season, following their pelagic larval phase). This is now defined as such in the first paragraph of the results (lines 57–58) and in the methods (line 282–283), and used consistently throughout the manuscript.

Comment 1.7: Line 194-196 – what does this mean ‘Loudspeaker playback matched that used in subsequent experimental trials’? The sound levels and spectra matched the healthy ecosystem? Then show the healthy system and the playback plotted together. This is a poorly worded and imprecise statement. The methods were identical but stimuli were not measured in each location? Also, you noted the bleaching impacted this system, so are you using impacted sounds? i.e., a poor soundscape for your treatment?

Response 1.7: This was meant to indicate that the playback methods were identical for both this initial test of loudspeaker performance and subsequent experimental loudspeaker deployments, not that the sound levels and spectra from loudspeaker playback matched the healthy ecosystem. The wording has been altered to explain this more precisely (lines 195–196).

For measurements of stimuli in experimental locations, see response 1.18.

Soundscapes used in the acoustic-enrichment treatment were taken from a healthy reef in the middle of the study site, recorded in November 2015. We have made it clear that this is before the two major bleaching events at the study site, which occurred in 2016 and 2017 (lines 259–261).

Comment 1.8: Later in this paragraph – So what were the depths at each site? This really affects sound transmission and sound levels.

Response 1.8: All experimental patch reefs were built in 2–4.5 m depth (mid-tide depth; tidal range during experiment ± 1.3 m), and there was no significant difference in depth between the three treatment groups. This information has been added to the methods (lines 236–238).

Comment 1.9: How was the Geospectrum particle motion data analyzed? This is not trivial? Sample rates? Was there a lowpass filter? Analyses window? Overlap?

Response 1.9: For both pressure and particle motion, recordings were taken with a sampling frequency of 48 kHz and analysed with a frequency range of 10–4000 Hz, an FFT size of 2048, a Hamming window and a 50% overlap. This information has been provided in the methods (lines 204–207).

Comment 1.10: Line 207 – ‘Analysis suggested....’ What analyses? Describe your methods. Sorry, this is very incomplete. Further, inaudible is not the right word here. Do you mean you could not distinguish the signal from the background noise? Note that in masking, temporal components of a signal play a huge roll. What was your integration time of analyses?

Response 1.10: The sentence in question (lines 208–209) is a topic sentence at the start of the paragraph; it summarises the findings of the analyses, with the rest of the paragraph (lines 209–223) going on to provide the necessary methodological detail. We have altered this topic sentence to summarise the paragraph more accurately, removing the vague term ‘analysis’ that was causing confusion.

We have changed “inaudible” to “could not distinguish a signal against the background noise floor” in all instances in this paragraph (lines 208 and 221; see also response 1.15).

Analyses were performed over three time frames: full-night comparisons at 1, 50 and 100 m; evenly-spaced repeated two-minute comparisons at 1, 50 and 100 m; and intermittent recordings at 50 m comparing ten-minute periods with the loudspeaker on with ten-minute periods with the loudspeaker off. These details are provided in lines 209–222 and in the caption for Figure 6.

Comment 1.11: Line 210 – what does ‘characteristically altered power spectra’ mean? Please be clear. This is not very precise. Also delete volume and used SPL or particle acceleration or received level.

Response 1.11: “Characteristically altered power spectra” refers to the visible differences in power spectral density plots (Fig. 6) – this has been made clear in the text (line 213). “Volume” has also been replaced by “sound-pressure (SPL) and particle-acceleration (PAL) levels” (line 212).

Comment 1.12: Line 212 – particle acceleration units are not very helpful or standard. See Casper and Mann papers, or Karlsen and Sand papers. Usually it is reported in m/s² or dB m/s but i guess it is because the differences are so small here (micrometers). Is that a realistic value? What is your variance?

Response 1.12: The values reported here are very similar to those in many other coral-reef acoustics studies by both our group and others (e.g. Nedelec *et al.* 2014 *Sci. Rep.*; Simpson *et al.* 2016 *Nat. Commun.*; Holmes *et al.* 2017 *J. of Exp. Mar. Biol. Ecol.*; Gordon *et al.* 2018 *PNAS*; McCormick *et al.* 2018 *Sci. Rep.*; McCormick *et al.* 2018 *Proc. R. Soc. B*). These papers all use the same units (dB re 1 μms^{-2}), which are those recommended by Nedelec *et al.* (2016 *Methods Ecol. Evol.*) in their introduction to particle motion in underwater acoustic ecology. Other units are used elsewhere, for instance in the measurement of louder sound sources such as whale song (Mooney *et al.* 2016 *Biol. Lett.*), but in the case of these relatively quiet sounds we prefer to match previous related studies with the current units.

Differences referred to in the text were simply read off the graphs in Figure 6, where full details of range and variance are provided in boxplots. As providing all of these details in-text would be cumbersome for the reader, we have limited these details to the figure, and have replaced the indicative values in the text with a direct reference to the figure (line 213).

Comment 1.13: Line 213 – again, volume is not a scientific word. You may be working in intensity or amplitude...

Response 1.13: As in response 1.11, this has been changed to SPL and PAL (lines 214–215 and 216).

Comment 1.14: Line 216 – what you testing here, full band SPL or certain bands?

Response 1.14: In all analyses, we were testing both SPL and PAL within the frequency band 10–4000 Hz, as the likely hearing range of young fishes. These details have been provided (lines 206–207).

Comment 1.15: Line 219 – Masking is a hearing question. Inaudible is not correct. Revise this. See work by J. Stanley and Richard Fay to properly assess what can and cannot be detected. This assumption needs to be revised.

Response 1.15: This has been changed accordingly, in lines 208–209 and 220–222 (also see response 1.10).

Comment 1.16: Line 251 – it is cleaner to play a silent treatment in case there is an electromagnetic signal of the system.

Response 1.16: This study was carried out to assess the impacts of acoustic enrichment of fish communities, rather than to understand the acoustic and/or electromagnetic drivers behind the observed differences. As such, we were not trying to understand which aspect of loudspeakers playing back reef sound was most attractive to reef fishes (acoustic or electromagnetic); just whether it enhanced recruitment relative to an increase in physical structure. This is why we used a physical structure control rather than a silent acoustic treatment control. Previous studies have compared the relative differences between reef sound and silent treatments, finding no effect of electromagnetic signals associated with silent controls, but this study was not concerned with such specific mechanical drivers of the observed effect. These points are both made in lines 135–142.

Comment 1.17: Line 262 – 1 m from surface but what was the depth (distance from the sound source)?

Response 1.17: The water depth was 3.5 m, falling within the range of depths at which experimental patch reefs were placed. This detail has been added (lines 267–268).

Comment 1.18: Line 271 – did you measure the received levels during the playback experiments? At least from a subset of your trials? Or at least calibrate each site? Measuring and monitoring your treatment is crucial.

Response 1.18: The pre-experimental test recordings were carried out at the deepest experimental reef in the study (4.5 m mid-tide depth). As sounds propagate less well in very shallow water, recordings at this reef represented a conservative estimate of acoustic isolation; acoustic halos around loudspeakers are likely to have been larger at this site than at any of the others in the experiment. This information has been added in lines 190–193.

Comment 1.19: Line 296 – did you test distance to nearest reef? Seem like an important parameter. Add if not.

Response 1.19: All experimental reefs were placed at a fixed distance of 25 m (as determined by GPS) from the nearest natural reef. This information is given in the caption for Figure 5 (line 184–185), and in the text of the methods (line 235).

Comment 1.20: Line 293 – what do believe your catch success is/was? Please report. This is a difficult job, catching these fish.

Response 1.20: This is indeed a difficult job, but the authors have decades of cumulative experience in catching these fishes at this field site, and learning from this wealth of experience was of great value to the observer (TACG) in carrying out this protocol. Further, the observer had a dive buddy who was constantly watching out for 'stray' fishes that attempted to escape across the

sand flat or burrow into the sand during the survey process. Finally, the dive buddy double-checked reefs for missed fishes after surveys were completed. To our knowledge, no fishes were missed in these surveys. This detail, along with information about the double-observing and double-checking systems have been included in the manuscript (lines 295–298).

Comment 1.21: Additional questions- where are you getting the rubble? From the beach, from near the site they were placed? All From one location?

Response 1.21: All rubble for all reefs in all three treatments was collected from a single location at the study site. This detail has been added to the methods text (line 238–239) and the location has been displayed on Figure 5.

Comment 1.22: Figure 1 – You mention there was an increase relative to the noon-moon/time of greatest recruitment. But also that there was a 10-day spread relative to treatment start date. Due to start-date-spread, it is not very easy to see the lunar affect, although this is referred to in the text. Can you plot the data relative to new-moon? Also, What does ‘abundance’ mean? Number of animals per site? ‘Raw data’ is not very explanatory... Suggest phrasing the x-axis of b or defining in the caption.

Response 1.22: We apologise for the confusion caused; we did not mean that we observed and quantified an increase relative to the new moon period, just that previous studies have found higher recruitment in new moon phases. As we don’t have data to confirm that this was the case in this instance, and plotting our data relative to the new moon is not possible due to the 10-day stagger, we have deleted this sentence. This paragraph (lines 68–79) now contains two (rather than three) reasons to suggest why asymptotic trajectories of damselfish abundance represent a stable dynamic equilibrium between recruitment and predation.

‘Abundance’ refers to total number of fishes per reef, and ‘raw data’ refers to unprocessed count data (as opposed to the model outputs presented in Figure 1a); these details have been clarified in the y-axis of Figure 1a, the x-axis of Figure 1b, the x-axis of Figure 2 and the caption to Figure 1 (lines 82 and 84).

Comment 1.23: Figure 4 – Delete this figure. It’s confusing. I know generally what you want to show but I don’t think it is represents what you want to show. For example, on the blue side, it looks like the recovered ecosystem is producing the low level sound. but really, it is more linear with increased sound leading to recruitment and recovery. I get the goal, but I don’t think it’s well devised.

Response 1.23: This figure has been modified to remove the confusion; the recovered ecosystem no longer points directly to the low level sound, as it did in our previous ‘in-line’ cycle style. Instead, we now present a ‘boxed’ alternative that makes each step in the process more distinct and the overall order of the schematic clearer. We feel that schematic explanations like this have significant explanatory power, especially in the context of journals with a wide readership like *Nature Communications*. We are happy to respect the views of the Editor on whether this clearer, revised figure should be included in the main MS (as now), in the Supplementary Material or not at all.

Comment 1.24: Figure 5 – Lizard Island seems incorrectly placed on mainland Australia. Note, it is off shore. Pls correct.

Response 1.24: This has been corrected.

Comment 1.25: Overall, What are the currents in the area? Were they measured? Did you see drop-off of recruitment based upon sites?

Response 1.25: Previous measurements of currents in this region of the Great Barrier Reef suggest that at this time of year they are predominantly wind-driven and therefore highly variable (Frith et al. 1986 Coral Reefs). We did not measure the currents in the area during this experiment; instead, to avoid any bias or confounds, we counterbalanced the assignment of reefs to the three experimental treatments (i.e. spatially adjacent reefs were never assigned to the same treatment). This is explained in lines 245–247, and shown in Figure 5.

Reviewer #2

Comment 2.1: My concern with the implications of using sound as a tool to recover fish populations (and I acknowledge that you also state “complementing existing local-scale reef management and restoration projects” in the conclusions) is that while the emitted sound might promise the fish a healthy environment to settle, it does not provide the essential resources that a healthy environment offers. Therefore, it attracts them to the reef under false pretences. Reef fishes are attracted to these sounds because they are cues for a suitable place to settle. Therefore, this methods may fool fish to settling into an environment that may not facilitate survival. While this study shows a higher abundance over the 40 days sampling period, it is unclear if higher recruitment is masking high mortality or if there are long term implications for settling onto a reef devoid of resources. This may be species specific also, whereby fish abundance and diversity increases, but the post-disturbance community is different to the pre-disturbance community. I understand that this is outside the scope of this study, but long term health and survivorship of species is important to fully understand if this method is a useful tool and therefore can facilitate recovery. Given the sound is only distinguishable at a close distance (<50m), it would appear that this method is not attracting them to the reef, rather it is influencing them to settle on the reef as they pass by. Tricking fishes to settle in a degraded habitat might be a death sentence, and it might be better to let these fish continue in the water column until they find a suitable habitat for settlement (in the case of patchy degradation). Fish communities change following a decline in reef healthy because it no longer provides resources for these fishes. Therefore, it is difficult to make the case that attracting fish through recruitment will recover fish populations if the environment is the same as when the fish declined. Aside from these two points, the authors have conducted an excellent study that is well presented and executed, and makes a valuable contribution to the field.

Response 2.1: We agree with Reviewer 2 that acoustic enrichment is unlikely to recover fish populations if the environment is the same as when the fish declined. This is why we originally stated “complementing existing local-scale reef management and restoration projects” in the conclusion. Acoustic enrichment is most likely to be effective as a tool if combined with other techniques that increase habitat suitability; we have made this point much more explicit in the conclusion (lines 159–162), and also added it to the abstract (lines 26–29). We also agree that the

functional composition of post-acoustic enrichment communities is important. As Reviewer 2 states, this is out of the scope of this experiment, but we have added a call for further research into this area (line 162–164).

Reviewers' Comments:

Reviewer #1:

Remarks to the Author:

Overall, mostly small modifications to the paper. I do have a few questions about Figure 6 (see below) and data recording, pre-stimulus. The idea and most work seem very good. Just making sure it's technically correct.

Ln 53 – change out keystone with important or vital, but keystone has a certain meaning. Not sure it's been defined for the family of your choice (or cite the paper that shows your family is keystone). (You note below Pomacentrids are chosen because 'visually surveyed accurately with minimal disturbance to the developing fish community' not because they are keystone.

Ln 89 – this is an intriguing result (the consistency) because many spp do not initially settle on reefs (see McCormick and Leaky, 1997; Nagelkerken 2002; Jaxion-Harm 2012; others). Such as Mullidae, which you measured. They settle off reefs to avoid predation, and move onto reefs at later life stages. You note predation at your reefs. You might also have such a large larval pool, that habitat is limiting and any reef protection is good. Your 'reefs' area also rubble (I think) so maybe that is 'off reef' for the fishes. You also used 'micro-reefs' within the larger reef structure, so different from the standard reef... I wonder what the authors think of this? Could they please add to a few sentences to the discussion? If you attracted larvae to the reef with smell, would you attract the whole community, as noted here? There is surely some literature on this using One Tree Island or similar models.

Ln 117 – I am not sure what the authors mean by 'increased detectability'. Could they please rephrase this phrase?

Ln 192 – This should be changed to read something like: Lower frequency (longer-wavelength) sounds tend to attenuate relatively rapidly in shallow water.

Figure 6C – Something is right with the acceleration measurements (or plot) below 1000 Hz. From 0-1000 they are decaying exponentially in dB which is already a log scale. It looks like there's a filter setting that is removing sounds below 1000 Hz. It is not in the pressure measurements, so not in the hydrophone recordings. But this should be sorted before publication. Below 1000 Hz (the frequencies of greatest sensitivity for most fish) the PA values presented are incorrect. The spectra data will affect the broadband PALs. In any case, that portion of the curve does not seem natural, nor a simple miscalculation.

Additionally, I am confused about the acceleration recordings. The authors note they recorded using a frequency sensitivity of 0-2 kHz but they plotted much higher (up to 4 kHz). How did they calculate the higher frequency data (2-4 kHz) without the higher sensitivity curves? And then they state all recordings were sampled at 48 kHz data. I am not sure most accelerometers work up to 48 kHz. They tend to be low frequency tools. Does this geospectrum system work up to 24 kHz (48 kHz sample rate)? Was the accelerometer system really recorded at 48 kHz, if so, did you apply a bandpass filter before the digital data acquisition? Otherwise the PA data are aliased. This may be a simple set of misstatements that need to be corrected but either the measurements or statements need to be corrected before publication.

In a slightly broader experimental context, the authors should show the levels measured at 25 m, the spacing of the reefs, to show the effect of one reef (acoustically) to another. Hopefully they did this.

The authors note they used a nocturnal playback (or several). This is great. Can they please provide the spectrograms of these in supplemental or similar? As well as a plot of SPL/SAL over vs. time? It

would be helpful to see the stimuli in a somewhat raw form and how it changed over the course of the night. And it would be helpful to compare that do the other playback treatments.

Reviewer #2:

Remarks to the Author:

I feel the authors have adequately addressed the comments and suggestions

Second response to reviewers, NCOMMS-19-01808A (Acoustic enrichment can enhance fish community development on degraded coral-reef habitat)

We are very grateful to both reviewers for their detailed, positive and constructive comments in both rounds of review of this manuscript. Below, we address the comments from Reviewer 1 that raise questions or require action; comments that were exclusively complimentary and/or summary statements (including all of Reviewer 2's response in this round) are excluded. We present original reviewer comments as *italicised text*, followed by our responses in **bold**. Line numbers refer to those in our revised manuscript, where all modified and referenced sections are highlighted in yellow.

Comment 1: Ln 53 – change out keystone with important or vital, but keystone has a certain meaning. Not sure it's been defined for the for family of your choice (or cite the paper that shows your family is keystone). (You note below Pomacentrids are chose because 'visually surveyed accurately with minimal disturbance to the developing fish community' not because they are keystone.

Response 1: This has been altered as suggested (line 52).

Comment 2: Ln 89 – this is an intriguing result (the consistency) because many spp do not initially settle on reefs (see McCormick and Leaky, 1997; Nagelkerken 2002; Jaxion-Harm 2012; others). Such as Mullidae, which you measured. They settle off reefs to avoid predation, and move onto reefs at later life stages. You note predation at your reefs. You might also have such a large larval pool, that habitat is limiting and any reef protection is good. Your 'reefs' area also rubble (I think) so maybe that is 'off reef' for the fishes. You also used 'micro-reefs' within the larger reef structure, so different from the standard reef... I wonder what the authors think of this? Could they please add to a few sentences to the discussion?

Response 2: We agree that the coral-rubble patch reefs in this experiment created marginal habitat on the edges of reefs, onto which a wide range of fish families settle; these include some fishes that are not traditionally associated with settling directly onto healthy reefs. We have added a sentence to reflect this (lines 245–247). We also agree that the findings in this study system would not necessarily be identical in other ecological contexts. This latter point is outside the remit of the current study, but as requested we have added a point to our discussion calling for further work addressing these important questions (lines 164–167).

Comment 3: If you attracted larvae to the reef with smell, would you attract the whole community, as noted here? There is surely some literature on this using One Tree Island or similar models.

Response 3: Although the use of olfactory cues to attract juvenile fish to reefs is an interesting question, it is outside the remit of this study. Current understanding of the role of smell in attracting larval fishes towards reefs is largely limited to studies carried out in laboratory-based flume choice tests (e.g. Dixon *et al.* 2008, *PRSB*; Munday *et al.* 2009, *PNAS*; Dixon *et al.* 2014, *Science*; Scott *et al.* 2016, *PRSB*). As the molecules involved in olfactory attraction, their synthesis and methods for releasing them at the reef scale are not yet fully established, field-based studies focussed on attracting larval fish with smell are currently less experimentally feasible than those concerning acoustic cues.

Comment 4: Ln 117 – I am not sure what the authors mean by ‘increased detectability’. Could they please rephrase this phrase?

Response 4: The meaning of this sentence has been clarified (lines 117–120).

Comment 5: Ln 192 – This should be changed to read something like: Lower frequency (longer-wavelength) sounds tend to attenuate relatively rapidly in shallow water.

Response 5: This has been changed as recommended (lines 195–196).

Comment 6: Figure 6C – Something is right with the acceleration measurements (or plot) below 1000 Hz. From 0-1000 they are decaying exponentially in dB which is already a log scale. It looks like there’s a filter setting that is removing sounds below 1000 Hz. It is not in the pressure measurements, so not in the hydrophone recordings. But this should be sorted before publication. Below 1000 Hz (the frequencies of greatest sensitivity for most fish) the PA values presented are incorrect. The spectra data will affect the broadband PALs. In any case, that portion of the curve does not seem natural, nor a simple miscalculation.

Response 6: We thank the reviewer for drawing our attention to the error in this figure. This was caused by a bug in the paPAM analysis package that we used to create the figure. We have subsequently corresponded with the authors of the paPAM package, who have revised the software to address the issue. The package now functions correctly and the issue has been resolved: we present corrected data in the PSD (Fig. 6C), the broadband PAL calculations (Fig. 6D) and the statistical comparisons of intermittent loudspeaker playback (lines 223–224). In all cases, the qualitative conclusions drawn from the data remain the same.

Comment 7: Additionally, I am confused about the acceleration recordings. The authors note they recorded using a frequency sensitivity of 0-2 kHz but they plotted much higher (up to 4 kHz). How did they calculate the higher frequency data (2-4 kHz) without the higher sensitivity curves? And then they state all recordings were sampled at 48 kHz data. I am not sure most accelerometers work up to 48 kHz. They tend to be low frequency tools. Does this geospectrum system work up to 24 kHz (48 kHz sample rate)? Was the accelerometer system really recorded at 48 kHz, if so, did you apply a bandpass filter before the digital data acquisition? Otherwise the PA data are aliased. This may be a simple set of misstatements that need to be corrected but either the measurements or statements need to be corrected before publication.

Response 7: We have checked the original manufacturer calibration data, and have discovered that this instrument is actually calibrated between 0 and 5 kHz. This has now been corrected in the manuscript (lines 205–206). The sampling rate of the recorder was 48 kHz, but all sounds were only analysed up to 4 kHz, as the likely upper limit of juvenile fish hearing (lines 209–210).

Comment 8: In a slightly broader experimental context, the authors should show the levels measured at 25 m, the spacing of the reefs, to show the effect of one reef (acoustically) to another. Hopefully they did this.

Response 8: The reefs were spaced at 100 m apart from one another (line 226–227). The recordings taken at 50 m and 100 m therefore demonstrate the effect of one reef’s playback on another (lines 215–226).

Comment 9: The authors note they used a nocturnal playback (or several). This is great. Can they please provide the spectrograms of these in supplemental or similar? As well as a plot of SPL/SAL over vs. time? It would be helpful to see the stimuli in a somewhat raw form and how it changed over the course of the night. And it would be helpful to compare that do the other playback treatments.

Response 9: As requested, a spectrogram and waveform of the nocturnal reef noise treatment has been provided in Supplementary Material (line 279, Fig. S1). The other treatments in the experiment did not involve playback (lines 248–263).